# The Effect of Single Bout Treatment of Heat or Cold Intervention on Delayed Onset Muscle Soreness Induced by Eccentric Contraction

**DOI:** 10.3390/healthcare10122556

**Published:** 2022-12-16

**Authors:** Riku Yoshida, Masatoshi Nakamura, Ryo Ikegami

**Affiliations:** 1Institute for Human Movement and Medical Sciences, Niigata University of Health and Welfare, 1398 Shimamicho, Kitaku, Niigata 950-3198, Japan; 2Faculty of Rehabilitation Sciences, Nishi Kyushu University, 4490-9 Ozaki, Kanzaki, Saga 842-8585, Japan

**Keywords:** lengthening muscle contraction, edema, treatment

## Abstract

We studied the preventive effects of heat or cold therapy after repeated eccentric contraction against torque reduction, muscle soreness, and range of motion (ROM) due to delayed-onset muscle soreness (DOMS). A total of 42 healthy male subjects were randomly allocated into three groups: the HEAT group received heat therapy using an ultra-short-wave device; the ICE group received ice therapy using an ice pack; the Control group received no intervention. The measurements included maximal voluntary isometric, concentric, and eccentric elbow flexion torque, elbow extension ROM, pressure pain threshold, and muscle soreness with stretching muscle thickness and echo intensity. The measurements were taken before (pre), after (post), after (t-post), one–four days after, and seven days after the muscle damage protocol. The results showed the main effect of time on all measurements, but no significant interactions were observed. The results of this study suggest that heat or cold therapy in the first 30 min after intense eccentric exercise is insufficient to exert a preventive effect against DOMS.

## 1. Introduction

Exercise with eccentric contraction (ECC) has been reported to induce delayed-onset muscle soreness (DOMS) [1,2,3]. A review by Clarkson et al. [1] demonstrated that DOMS causes maximal muscle strength reduction, local edema, and muscle soreness. It is important to prevent or reduce DOMS, which may lower workout motivation. Furthermore, resistance training (RT) with ECC has been shown to be more effective in increasing muscle strength and hypertrophy than RT with concentric contraction (CON) [4,5,6]. In addition, a recent study reported that ECC-only RT is as effective as RT with a combination of CON and ECC [7]. Thus, to promote RT with ECC, it is necessary to establish methods for preventing and reducing the problems associated with DOMS.

Clinically, first aid for muscle injuries follows the RICE (Rest, Ice, Compression, and Elevation) principle [8]. As represented by the RICE protocol, cold therapy has long been employed to treat musculoskeletal pain, presuming that acute inflammation is the cause of DOMS [9]. Cold therapy has physiological effects, such as decreased local metabolism, capillary osmotic pressure, sensory impulses to the center due to delayed stimulus transmission, muscle spindle activity, vasoconstriction and subsequent dilation, and increased sensory receptor thresholds [10]. These actions are expected to alleviate inflammation and edema, improve blood circulation, provide analgesia, and reduce muscle spasms. However, recent studies, mainly in animal experiments, have reported that cold therapy delayed muscle regeneration after muscle injury induction [11] and that cold therapy immediately after severe muscle injury inhibited muscle regeneration [12]. Although some reports suggest that cold therapy inhibits muscle regeneration [12,13], it is recommended by guidelines [14]. Therefore, the effect of cold therapy on DOMS prevention still needs further investigation. In a human study, Yamane et al. investigated the effects of regular post-exercise cold application on muscular and vascular adaptations induced by moderate-intensity RT. The results indicated that regular post-exercise cold application to muscles might attenuate muscular and vascular adaptations to RT [15]. Thus, it was suggested that cooling after the training intervention inhibits long-term muscle adaptation and other factors. However, Doungkulsa et al. [16] performed 20 min (four sessions × 5 min) of air-pulsed cryotherapy on DOMS-induced elbow flexor muscles for five consecutive days and found improvement in pressure pain threshold (PPT), range of motion (ROM), muscle soreness at stretching (SOR-st), and brachial circumference. Thus, post-exercise cold therapy could have a negative effect on long-term muscle adaptation, but a preventive effect against DOMS can be expected. However, to date, there are no studies on using ice packs, which are frequently used in sports and rehabilitation settings, and the effect of cold therapy immediately after exercise is unknown.

For hyperthermia, physical therapy includes using hot packs, paraffin baths, microwaves, and ultrasound techniques. In the inflammatory phase, hyperthermia is not advised as it increases blood flow, which heightens discomfort and swelling. On the other hand, it has been reported that the increase in muscle temperature by hyperthermia treatment induces heat shock proteins (HSP), dilates peripheral blood vessels, and stimulates inflammation and that hyperthermia treatment using ultra-short waves increases local blood flow, which may increase energy supply and contribute to the initiation of the repair and injury processes [17]. Saga et al. [18] performed heat treatment on elbow flexor muscles one day before an intense exercise in adult males. The results indicated that heat preconditioning, applied one day prior to ECC1, suppressed the decrease in maximum voluntary contraction and ROM. However, this was a pre-exercise intervention, and the recovery process of DOMS symptoms following a similar intervention after exercise has been unclear. It is also unclear which intervention method is more effective, cold or heat therapy. Therefore, through this comparison, an effective cold or heat therapy can be established to solve the problems of DOMS caused by ECC.

This study aimed to investigate the time course changes in the elbow flexion torque, elbow extension ROM, PPT, SOR-st, muscle thickness (MT), and muscle echo intensity (EI) after a single bout of heat or cold therapy following intense ECC exercise. The hypothesis of this study was that heat or cold therapy could improve each measurement and effectively prevent DOMS symptoms.

## 2. Materials and Methods

### 2.1. Participants

The participants were 42 healthy male adults. Since the previous study [19] reported that DOMS is more likely to occur in males compared to females, only males were recruited as participants in the study. Thus, the inclusion criteria were male university students who had not received resistance training within the previous two months. Exclusion criteria were skeletal muscle disorders, upper-limb or shoulder joint disorders, or neuromuscular diseases. The participants were instructed to avoid icing, heat, and strenuous exercise of the upper extremities for physical therapy during the experimental period. Furthermore, each participant was prohibited from taking a bath and was instructed to minimize showering during the experimental period. Before the start of the experiment, the participants were informed orally and in writing of the contents and points to be considered and provided written consent to participate in the experiment. This study was conducted in accordance with the rules of the Ethics Committee of Niigata University of Health and Welfare. The purpose of the study and possible risks were fully explained to the participants in advance. We calculated the sample size required for repeated two-way analysis of variance (ANOVA) (effect size = 0.40, alpha error = 0.05, power = 0.85) using the G*power software (version 3.1, Heinrich Heine University, Düsseldorf, Germany). The required number of participants for this study was 14 in each group.

### 2.2. Study Design

A total of 42 sedentary subjects were randomly divided into three groups of 14 each after DOMS induction. A randomization sequence was created using software (Microsoft Office Excel 2007, Microsoft, Redmond, WN, USA), and a computer-generated random list was used for allocation. Additionally, participants, therapists, and the examiner were blindfolded at the time of data measurement.

### 2.3. Study Protocol

Figure 1 presents the experimental protocol. Each participant was observed in the laboratory for five consecutive days and again after two days. Before the procedure, the participants’ anthropometric characteristics were measured. Subsequently, the baseline assessment (pre) of outcomes, muscle damage, DOMS elicitation tasks (muscle damage protocol), and assessments immediately (post) and 30 min after the muscle damage protocol (t-post) were conducted on the first day. The data were recorded from September 2021 to June 2022, and the recording location was a university laboratory. The first session took about 70 min, and the other sessions took about 10 min. Thus, the duration of the first and the second session were different. Follow-up visits were made at 24 h (day 1), 48 h (day 2), 72 h (day 3), 96 h (day 4), and 168 h (day 7) after the muscle damage protocol associated with delayed effects. These measurements consisted of assessments of muscle function [muscle pain, pain threshold, MT, and maximum voluntary isometric (MVC-ISO), shortening (MVC-CON), and lengthening (MVC-ECC) contractions].

### 2.4. Procedures

#### 2.4.1. Eccentric Exercise-Induced Muscle Damage Protocol

The muscle damage protocol was performed using an isokinetic dynamometer (Biodex System 3, New York, NY, USA) to induce muscle damage in the elbow flexor muscles of the dominant arm. Before the muscle damage protocol, the participants were familiarized with ECC. The habituation exercise consisted of one ECC below the lower limit of maximum elbow joint flexion. Before each repetition, the participant’s elbow joint was positioned at 10° of flexion. The participants were instructed to perform a 90° ROM (10°–100° elbow flexion) at an angular velocity of 60°/s while returning the elbow to 10° extension with an isokinetic dynamometer resistance [20]. The muscle damage protocol was initiated immediately after the habituation exercise, and 30 consecutive maximal ECCs of elbow flexion were performed. The same angular velocity and ROM as in the familiarization exercise were used with an 18-s rest between contractions.

#### 2.4.2. Elbow Joint Extension ROM

According to previous studies [16,21], ROM was measured using a goniometer in the back-lying position in an anatomically correct posture, and joint angles were measured from the relaxed position with the arms along the body to the maximum extension position. Measurements were taken before and after the muscle damage protocol and on days 1–4 and 7 after the muscle damage protocol. A single measurement was employed for the analysis.

#### 2.4.3. Muscle Soreness and PPT

According to previous studies [16,18,22], using a visual analog scale that had a 100-mm continuous line with “not sore at all” on one side (0 mm) and “very, very sore” on the other side (100 mm), the magnitude of elbow flexor muscle soreness was assessed via muscle stretching at the elbow extension ROM measurement. Muscle soreness during ROM measurement was gathered twice to determine soreness during stretching, and the average value was used for further analysis.

PPT measurements were performed using an algometer (NEUTONE TAM-22 (BT10); TRY ALL, Chiba, Japan) in the supine position. The measurement sites were 50% (PPT50), 60% (PPT60), and 70% (PPT70) of the lateral epicondyle of the humerus from the acromion. Each participant lay on a bed in the supine position, with the relaxed dominant arm at the side and the forearm supinated. With continuously increasing pressure, the metal rod of the algometer was used to compress the soft tissue in the measurement area. The participants were instructed to immediately press a trigger when pain, rather than just pressure, was felt. The value read from the device at this time point (kilograms per square centimeter) corresponded to the PPT. The average value was used for further analysis.

#### 2.4.4. MVC Torque

The MVC torque was measured at 90° (MVC-ISO) of elbow flexion in the same setting as the training using the isokinetic dynamometer [21]. Each contraction lasted for 3 s; two measurements were taken for each angle with a 45-s interval, and the larger value of the two measures was used for further analysis. After the MVC-ISO torque measurement, the MVC torque of concentric and ECC of the elbow flexors was measured in the same setting of the dynamometer as the MVC-ISO torque measures. The MVC-CON torque was measured at 60°/s and the MVC-ECC torque at 60°/s in this order [23]. The rest time between measurements was 60 s. The ROM was 100° for the MVC-CON and MVC-ECC torque measurements; the starting angle was 10° for MVC-CON, and 100° elbow flexion for MVC-ECC, where the fully extended elbow joint was defined as 0°. In the case of the MVC-CON torque measurement, each participant was instructed to perform MVC from 10° to 100° of elbow flexion only, and the arm was passively returned to the starting angle (10° elbow flexion) while relaxed. For the MVC-ECC torque measurement, after performing eccentric MVC from 100° to 10° of elbow flexion, the arm was passively returned to the starting angle (100° elbow flexion). In the MVC-CON torque measurement, each participant performed MVC three times consecutively with a 60-s rest between contractions, and the maximum torque obtained was used for the subsequent analysis. This was also the case for the MVC-ECC measurement. During all measurements, verbal encouragement was provided to the participants. The torque of each contraction was monitored and recorded using an analog-to-digital converter (PowerLab 8/30, AD Instruments, Colorado Springs, CO, USA) connected to a personal computer with the analysis software (Lab Chart 7, AD Instruments).

#### 2.4.5. MT and EI

Referring to Radaelli et al. [24], the biceps brachii and brachialis MT of the dominant arm were measured via B-mode ultrasonography using an 8-MHz linear probe (LOGIQ e V2; GE Healthcare Japan, Tokyo, Japan). The ultrasound intensity was 78.0, frequency 8.0 MHz, and depth 6.0 cm—consistent overall measurements at different time points across the participants. The investigator minimized the pressure of the probe against the skin as much as possible, and the same investigator took all measurements. The measurement sites were 50% (MT50), 60% (MT60), and 70% (MT70) of the lateral epicondyle of the humerus from the acromion. Each participant lay on a bed in the supine position with a relaxed dominant arm at the side and the forearm supinated. Ultrasound measurements of the transverse axis were repeated twice, and the MT of the biceps brachii and brachialis was measured as the distance from the inner edge of the fascia to the humerus.

EIs were determined via computer-assisted 8-bit gray-scale analysis using the standard histogram function in the ImageJ software (National Institute of Health, Bethesda, Rockville, MD, USA, version 1.37). The measurement sites were 50% (EI50), 60% (EI60), and 70% (EI70) of the lateral epicondyle of the humerus from the acromion. The regions of interest that included the elbow flexor muscles but avoided the surrounding fascia were selected. The mean EI of the regions was expressed as a value between 0 (black) and 255 (white). A single measurement was used for the analysis.

#### 2.4.6. Treatment

Based on previous studies [12,25], immediately after the muscle damage protocol, heat or cold therapy was administered to the elbow flexor muscles in the HEAT and ICE groups, respectively. A microwave therapy device (Microwave Therapy ME-7250, OG Wellness Technologies Co., Ltd. Okayama, Japan) was used for the thermal treatment of the elbow flexor muscles. Immediately after the muscle damage protocol, the participants’ elbow flexor muscles were irradiated with microwaves at 80 W for 30 min from approximately 15 cm. For cold therapy, the participant’s elbow was placed on a pillow, and the elbow flexor muscles were immobilized with the ice bag filled with ice cubes (icing bag size M, Mizuno Corp, Tokyo, Japan) for 30 min. For the CONT group, the patients remained supine with the shoulder joint in 90° abduction, 45° external rotation, and the elbow joint in extension for 30 min.

#### 2.4.7. Statistical Analysis

The graph creation, calculations, and statistical analyses were performed using GraphPad Prism version 9.0 for Windows (GraphPad Software, San Diego, CA, USA). Data normality was assessed using the Shapiro–Wilk test. Repeated two-way ANOVA with two factors (group: CONT vs. HEAT vs. ICE, time: pre vs. post vs. t-post vs. day 1 vs. day 2 vs. day 3 vs. day 4 vs. day 7 after the muscle damage protocol) was employed to compare the groups in terms of the changes in the dependent variables. The magnitude of change in each variable from pre to day 7 was compared using the Bonferroni correction. The differences were considered statistically significant at an alpha level of 0.05. Descriptive data were expressed as mean ± SD.

## 3. Results

Repeated measures via two-way ANOVA were performed in this study, and no significant interactions were found for all measures. For this reason, we examine the data of all groups together as a post hoc test.

### 3.1. Participants

The control group (CONT; age 20.7 ± 0.7 years, height 172.6 ± 3.7 cm, weight 64.5 ± 6.8 kg) rested for 30 min after DOMS induction; the Heat group (HEAT; age 20.8 ± 0.9 years, height 173.0 ± 4.7 cm, weight 61.0 ± 4.5 kg), received heat therapy; the Ice group (ICE; age 21.0 ± 0.5 years, height 172.2 ± 3.2 cm, weight 62.9 ± 6.4 kg), received cold treatment on the elbow flexor muscles during 30 min of bed rest after DOMS induction. No significant differences were observed in physical characteristics between the groups.

### 3.2. Muscle Temperature

In a pilot study, we measured muscle temperature in five healthy males with a surface-type deep body thermometer (Core temp CTM-210; Telmo, Tokyo, Japan). The results indicated that the muscle temperature increased from 34.2 °C ± 0.6 °C (mean ± SD) and 34.3 °C ± 0.9 °C (mean ± SD) before intervention to 39.1 °C ± 0.5 °C (mean ± SD) and 15.8 °C ± 2.1 °C (mean ± SD) after heat or cold therapy in the HEAT and ICE groups, respectively (Figure 2).

### 3.3. Muscle Damage Protocol

Figure 3 presents the main effect of time over 30 maximal ECC torque of the elbow flexor in the muscle damage protocol (F = 102.78; *p* < 0.01). However, no significant interaction was observed between factors (F = 0.85; *p* = 0.76).

### 3.4. Elbow Extension ROM

For the elbow extension ROM, no significant interaction was observed between factors (F = 1.20; *p* = 0.27), but the main effect of time was found (F = 10.05; *p* < 0.01, Figure 4B). The post hoc tests revealed significant (*p* < 0.05) decreases on days one, two, and three compared with pre. Furthermore, a significant increase was observed on day seven compared with days one, two, three, and four.

### 3.5. Muscle Soreness and Pain Pressure Threshold

For the SOR-st, no significant interactions were observed between factors (F = 1.17; *p* = 0.29). However, the main effect of time was observed (F = 32.87; *p* < 0.01, Figure 4A). The post hoc tests revealed a significant (*p* < 0.05) increase on days one, two, three, four, and seven compared with pre. Significant increases were observed on days one, two, three, four, and seven compared with post and on days one, two, three, and four compared with t-post for SOR-st. Furthermore, significant increases were observed on days two and three compared with day one for SOR-st. On the other hand, a significant decrease was observed on day seven compared with days two, three, and four.

For PPT50, PPT60, and PPT70 (Figure 5), no significant interactions between factors were observed (F = 1.59; *p* = 0.07, F = 1.52; *p* = 0.10, F = 1.51; *p* = 0.10, respectively). However, the main effect of time was observed (F = 35.17; *p* < 0.01, F = 32.87; *p <* 0.01, F = 15.11; *p <* 0.01, respectively). The post hoc tests revealed that significant decreases were observed on days one, two, and three compared with pre, post, and t-post for PPT50. Significant increases were also observed on days four and seven compared with days one, two, and seven compared with pre, post, t-post, and days one, two, three, and four for PPT50. The post hoc tests revealed that significant decreases were observed on days one, two, and three compared with pre, post, and t-post for PPT60. Furthermore, significant increases were observed on days four and seven compared with days one, two, three, and seven compared with pre, post, t-post, and days one, two, three, and four for PPT60. The post hoc tests revealed a significant decrease on days one and two compared with t-post and on days one, two, and three compared with t-post for PPT70. In addition, significant increases were observed on days four and seven compared with days one, two, three, and seven compared with pre, post, and t-post for PPT70.

### 3.6. MVC Torque

For the MVC-ISO (Figure 6A), MVC-CON (Figure 6B), and MVC-ECC (Figure 6C), no significant interactions were observed between factors (F = 0.20; *p* = 0.99, F = 0.63; *p* = 0.80, F = 0.47; *p* = 0.92, respectively). However, the main effects of time were observed (F = 153.50; *p* < 0.01, F = 90.06; *p* < 0.01, F = 168.30; *p* < 0.01, respectively).

The post hoc tests revealed significant decreases on post, t-post, and days one, two, three, four, and seven compared with pre for MVC-ISO, MVC-CON, and MVC-ECC. A significant increase was observed on day four compared with t-post and days one, two, three, and seven compared with post, t-post, and days one, two, three, and four for MVC-ISO. Moreover, a significant decrease was observed on days one and two compared with post for MVC-CON. A significant increase was also observed on day four compared with days two and seven compared with t-post and days one, two, three, and four for MVC-CON. A significant decrease was observed on t-post and days one and two compared with post for MVC-ECC. In addition, a significant increase was observed on day seven compared with post, t-post, and days three and four and on days four and seven compared with days one and two for MVC-ECC.

### 3.7. MT and EI

For MT50, MT60, and MT70, no significant interactions between factors were observed (F = 0.88; *p* = 0.57, F = 0.54; *p* = 0.90, F = 0.97; *p* = 0.48). However, the main effect of time was observed (F = 20.70; *p* < 0.01, F = 34.30; *p* < 0.01, F = 43.05; *p* < 0.01, Figure 7). The post hoc tests revealed that significant increases were observed on post, t-post, and days one, two, three, four, and seven compared with pre for MT60 and MT70. On the other hand, a significant decrease was observed on day seven compared with days three and four for MT60. Significant increases were also observed on days one, two, and three compared with post and on days one, two, three, and four compared with t-post. No significant difference was observed for MT50.

For EI50, EI60, and EI70, no significant interactions between factors were observed (F = 0.91; *p* = 0.54, F = 0.70; *p* = 0.76, F = 1.33; *p* = 0.18). However, the main effect of time was observed (F = 17.21; *p* < 0.01, F = 12.98; *p* < 0.01, F = 15.72; *p* < 0.01, Figure 8). The post hoc tests showed that significant increases were observed on post, t-post, days two, three, four, and seven compared to pre for EI50 and EI60. A significant increase was observed on day four compared with post and on days four and seven compared with day one for EI50. In addition, significant increases were observed on days four and seven compared with day two and on day four compared with day three for EI50. Furthermore, significant increases were observed on day seven compared with post, t-post, and days two, four, and seven compared with day one for EI60. Significant increases were observed on post, t-post, and days one, two, three, four, and seven compared with pre for EI70. In addition, a significant increase was observed on day seven compared with t-post and on days one, two, four, and seven compared with post for EI70.

## 4. Discussion

In this study, 30 min of heat or cold therapy immediately after repeated ECC was performed. The results indicated that neither the HEAT nor the ICE group exhibited improved elbow flexion strength, ROM, SOR-st, PPT, MT, or EI compared with the CONT group. These results indicate that 30 min of heat or cold therapy immediately after intense ECC exercise does not improve DOMS symptoms. Wang et al. [26] reported in their review that heat or cold therapy within 24 h after DOMS induction reduces DOMS-induced pain. However, in their review, they reported different results depending on the intervention, temperature, and treatment time. Therefore, the results of this study suggest that heat or cold therapy may not be effective in preventing DOMS.

### 4.1. Effect of the HEAT Therapy

The following factors were thought to be responsible for the preventive effect of post-exercise hyperthermia against DOMS: (1) increased intramuscular blood flow, oxygen supply to the injured area, accelerated clearance of inflammatory factors through tissue metabolism promotion [27], and (2) the effect of muscle repair by HSP, especially HSP72, as a muscle protein that suppresses muscle protein synthesis and degradation. However, in the present study, 30 min of heat therapy for DOMS in the elbow flexor muscles elicited by ECC did not exert a significant preventive effect against DOMS.

In a previous study, calcium homeostasis was altered, and intracellular calcium concentrations increased when muscle injury occurred. In the human skeletal muscle, it has been reported that nitric oxide (NO) production is significantly increased during muscle soreness, which may inhibit muscle force generation [28]. The co-occurrence of these events induces inflammation and muscle pain and impairs muscle function. Therefore, processes related to this mechanism need to be controlled to prevent and/or reduce muscle damage, muscle pain, and loss of muscle function. HSP72 induced by heat treatment has been reported to protect the E–C coupling structure and reduce toxicity in NO [29,30]. A previous study demonstrated that thermal stimulation the day before exercise suppressed the decrease in muscle strength and ROM caused by DOMS [18]. It was considered that the HSP72 expression was involved in this result. In addition, the acquisition of stress tolerance in HSP was taken into account. The involvement of stress-induced HSP in stress tolerance has been reported in many previous studies [31,32]. Theodorakis et al. [33] demonstrated that once cells become HSP-induced and tolerant to stress in human experiments, HSP70 mRNA transcription decreases, and HSP induction is suppressed after the subsequent stress. Therefore, it is possible that the effect of HSP induction by thermal stress itself, given immediately after the DOMS-induced protocol, as in the present study, was attenuated by DOMS induction.

Furthermore, the frequency of heat therapy may have affected the recovery of DOMS in the HEAT group. Oishi et al. [34] induced heat stress in rats by immersing the lower body in the water at 42 °C ± 1 °C for 30 min every other day for two weeks. The results indicated that the HSP72 values were greater in the heat stress-induced rats than in the CONT group. Therefore, it was suggested that heat therapy frequency is important in inducing HSP. In addition, the lower power of the heat therapy used in this experiment than in previous studies [18] may have resulted in insufficient HSP72 induction or removal of pain substances produced by DOMS due to increased blood flow. Thus, it is considered that the treatment effect of DOMS in the HEAT group in this study could not be obtained.

### 4.2. Effect of the ICE Therapy

In cold therapy, cold stimulation has been reported to suppress metabolism and inflammation by constricting local blood vessels [35]. The guidelines for treating acute tissue injury recommend complete cryotherapy for approximately 15–20 min several times a day until the swelling resolves within 72 h after injury [14]. On the other hand, Kawashima et al. [12] reported that cold therapy inhibits macrophage accumulation and delays muscle regeneration after muscle injury. In this study, the effect of the 30-min cold therapy on DOMS in the elbow flexor muscles elicited by ECC was also examined and did not exert a significant preventive effect against DOMS. Therefore, the results indicated that a single cold therapy does not promote exacerbation and recovery effects of DOMS in human studies. These results may have been related to these factors: (1) the magnitude of the muscle damage and (2) the frequency and duration of cold therapy.

Regarding the magnitude of muscle damage, Nosaka et al. [36] performed 12 or 24 MVC-ECC repetitions at the elbow flexor muscles. The results indicated that the group that performed 24 MVC-ECC repetitions had a more significant decrease in ROM and an increase in upper arm circumference, which are symptoms of DOMS, than the group that performed 12 MVC-ECC repetitions. The muscle damage protocol used in this study was conducted with 30 repetitions of MVC-ECC; therefore, we believe that it induced severe DOMS. A previous study has also demonstrated that cold therapy in severe DOMS did not affect each measurement. Paddon et al. [37] performed 110% 1RM dumbbell curls at eight repetitions × eight sets of elbow flexion with ECC only, followed by five repetitions of 20-min immersions in 5 °C cold water with a 60-min break on the side on which the exercise was performed. They reported no change in MVC-ISO reduction and no pain during contraction immediately after DOMS induction up to four days later. Furthermore, Paddon et al. [37] found that MVC-ISO did not return to the pre-values until four days later. In this study, MVC-ISO did not return to the pre-values on days four or seven. Therefore, regardless of the magnitude of muscle damage, we believe that cold therapy did not affect the accumulation of macrophages or inhibit muscle regeneration, as reported in animal studies [13].

The frequency and duration of cold therapy were cited as reasons for the lack of preventive effects against DOMS. Doungkulsa et al. [16] reported that air-pulse cryotherapy (−30 °C) was performed for five consecutive days at 20-min sessions (four sessions × 5 min) after DOMS induction on the elbow flexor muscles. The results indicated a significant improvement air-pulse cryotherapy group in SOR-st, PPT, and upper arm circumference compared with the CONT group. However, Kawashima et al. [12] found that rats under anesthesia with icing (30 min × three sets, every 2 h) immediately after MVC-ECC, 24 h, and 48 h were DOMS-induced. In that study, macrophage accumulation was suppressed, and muscle regeneration was inhibited. Therefore, it is suggested that high-frequency and prolonged cold therapy inhibits muscle regeneration, but in the present experiment on human subjects, no recovery or adverse effects on DOMS were observed as only a single shot of cold therapy was applied.

### 4.3. Limitations

This study has four limitations. First, the target muscles in this study were elbow flexor muscles. In the future, it is necessary to examine the effects of heat and cold therapy on DOMS in the trunk and lower-limb muscles. Second, the study subject was a sedentary young adult male. It is necessary to examine the effects of this study on people with a history of RT and the elderly. Third, the therapeutic effect differed depending on the timing of both interventions. Therefore, it is necessary to investigate the treatment effect by changing the timing of intervention, such as before or one day after DOMS induction. Fourth, the duration of both interventions was 30 min. In future studies, it is necessary to change the treatment duration because of the possibility of a capacity–response relationship in the intervention. Fifth, a single intervention session was performed with heat or cold applications, so it is necessary for future research to evaluate the effects of multiple sessions over a longer period. Sixth, the present study’s sample size calculations ensured that the study was sufficiently powered; further studies with larger sample sizes are needed to corroborate your findings.

## 5. Conclusions

Our results suggest that a 30-min heat or cold therapy administered immediately after DOMS induction in the elbow flexor muscles does not suppress DOMS. However, the effects of heat or cold therapy may vary with multiple treatments, different times, and different temperatures.

## Figures and Tables

**Figure 1 healthcare-10-02556-f001:**
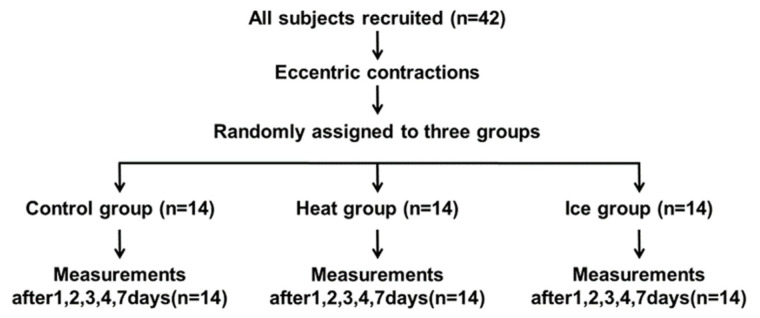
Experimental design and protocol. A total of 42 healthy university students were allocated to one of the three treatment groups; in the Control group, the participants rested for 30 min; in the HEAT group, the participants underwent 30-min microwave irradiation at 80 W from a distance of about 15 cm; in the ICE group, the participant’s elbow was placed on a pillow, and the elbow flexor muscles were immobilized with the icebag for 30 min (*n* = 14 per group)–all groups were placed in the supine position with the shoulder joint in 90° abduction and 45° external rotation and the elbow joint in extension during treatment. The elbow joint was placed in extension. All groups underwent various measures before and after the muscle damage protocol, after treatment, and 1–4 and 7 days after the muscle damage protocol.

**Figure 2 healthcare-10-02556-f002:**
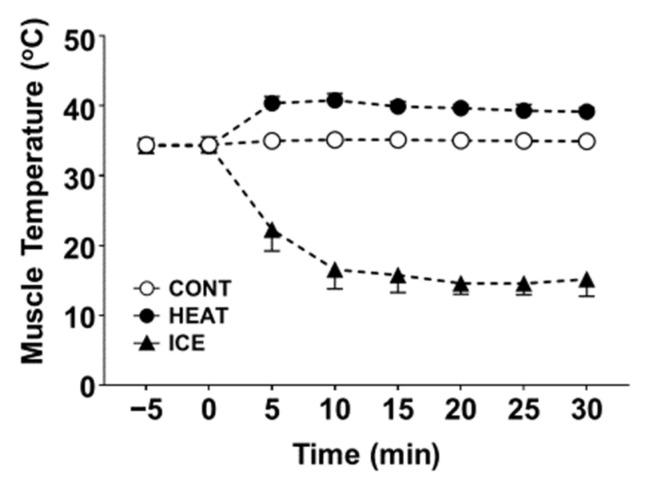
Mean of muscle temperature in each group.

**Figure 3 healthcare-10-02556-f003:**
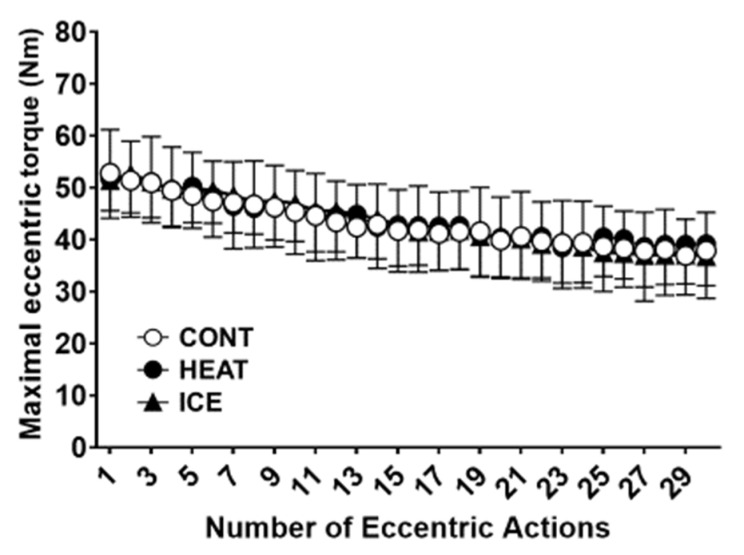
Changes in the elbow flexion torque (mean ± SD of 14 participants) over 30 maximal voluntary eccentric contractions.

**Figure 4 healthcare-10-02556-f004:**
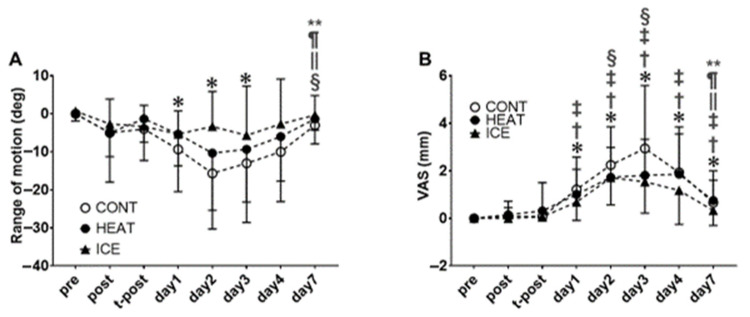
Changes in elbow extension range of motion (**A**) and muscle soreness at stretching by visual analog scale (**B**). * significant compared with pre (*p* < 0.05); †, significant compared with post; ‡, significant compared with t-post; §, significant compared with day 1; ||, significant compared with day 2; ¶, significant compared with day 3; **, significant compared with day 4.

**Figure 5 healthcare-10-02556-f005:**
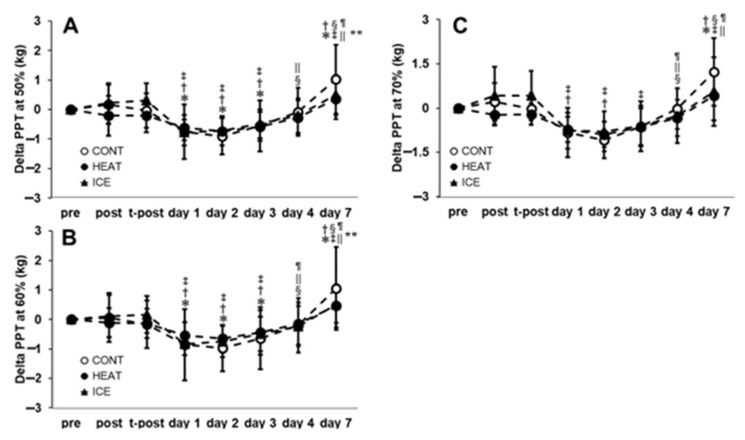
Changes in pain pressure thresholds at 50% (**A**), 60% (**B**), and 70% (**C**) of the upper arm. * significant compared with pre (*p* < 0.05); †, significant compared with post; ‡, significant compared with t-post; §, significant compared with day 1; ||, significant compared with day 2; ¶, significant compared with day 3; **, significant compared with day 4.

**Figure 6 healthcare-10-02556-f006:**
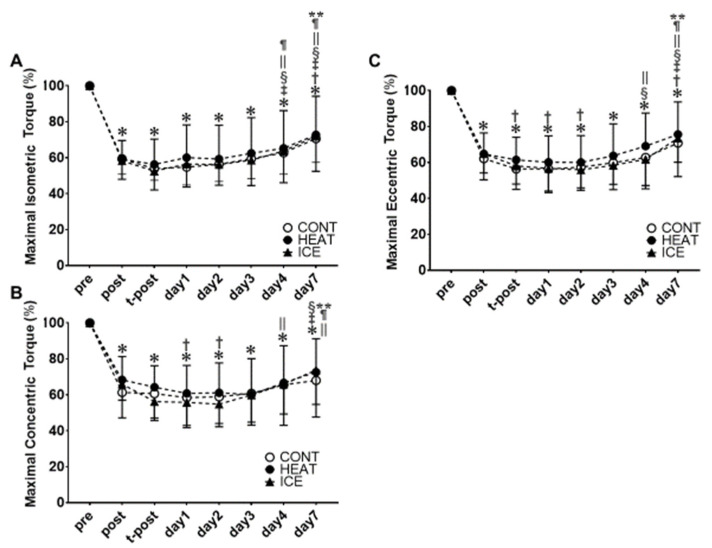
Changes in the maximal voluntary contraction (MVC) torque at isometric (**A**), concentric (**B**), and eccentric (**C**) elbow flexion torque. *, significant compared with pre (*p* < 0.05); †, significant compared with post; ‡, significant compared with t-post; §, significant compared with day 1; ||, significant compared with day 2; ¶, significant compared with day 3; **, Significant compared with day 4.

**Figure 7 healthcare-10-02556-f007:**
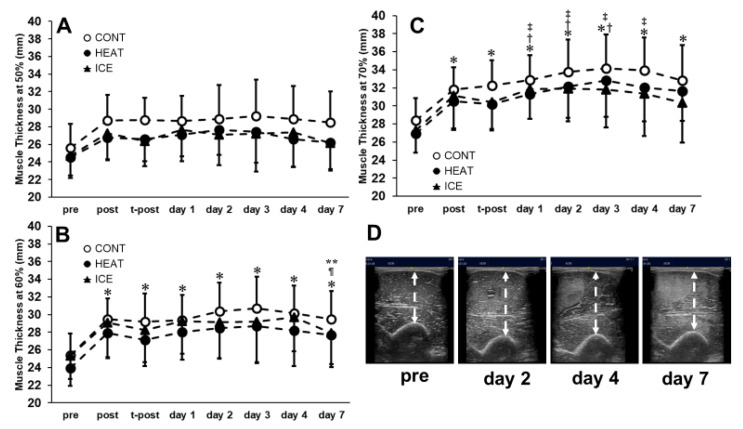
Changes in muscle thickness at 50% (**A**), 60% (**B**), and 70% (**C**) of the upper arm. Changes in muscle thickness and echo intensity of elbow flexor muscles after muscle damage protocol. (**D**) *, Significant compared with pre (*p* < 0.05); †, significant compared with post; ‡, significant compared with t-post; ¶, significant compared with day 3; **, significant compared with day 4.

**Figure 8 healthcare-10-02556-f008:**
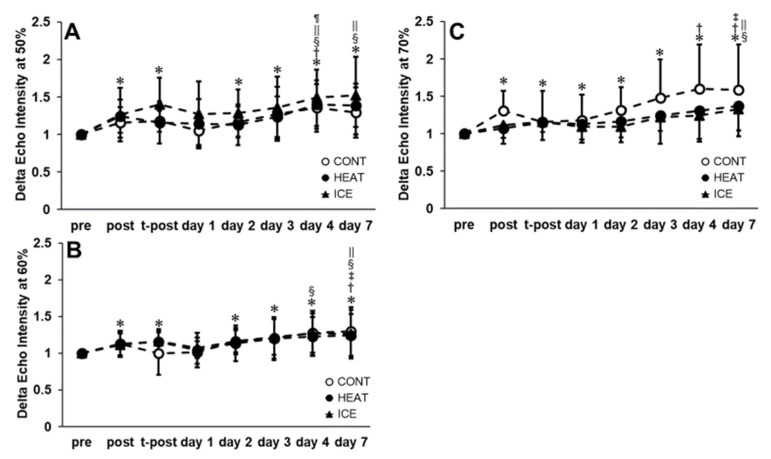
Changes in echo intensity at 50% (**A**), 60% (**B**), and 70% (**C**) of the upper arm. *, Significant compared with pre (*p* < 0.05); †, significant compared with post; ‡, significant compared with t-post; §, significant compared with day 1; ||, significant compared with day 2; ¶, significant compared with day 3.

## Data Availability

All data supporting the conclusions of this study will be fully pro-vided upon request by the authors.

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
