# Peer review of "The Effect of Single Bout Treatment of Heat or Cold Intervention on Delayed Onset Muscle Soreness Induced by Eccentric Contraction"

_healthcare, 2022, doi:10.3390/healthcare10122556_

Round 1

Reviewer 1 Report

healthcare-2080973_review

Title: The treatment effect of heat or cold intervention on delayed-onsetmuscle soreness induced by eccentric contraction

Comments for Authors

Dear authors,

I have carefully read your paper, which investigated the effects of a single session of heat or cold therapy applied on the elbow flexor muscles after repeated eccentric contraction on torque reduction, muscle soreness, and range of motion due to delayed onset muscle soreness in healthy male participants.

In your results observed the effect of time on the measurements, but no significant interactions were obtained. Therefore, it appears that heat or cold therapy applied in the first 30 min after intense eccentric exercise is insufficient to exert a preventive effect against delayed onset muscle soreness.

In general, the manuscript is well-written. The text is understandable and organized. However, I found some issues in introduction, materials and methods, results, discussion and conclusion sections that should be addressed to improve the paper, in my opinion.

Specific comments:

Introduction

-  Page 1, line 35. The RICE (rest, ice, compression, and elevation) protocol is known to most, but there may be any readers who are not familiar with this term, please add the full meaning of the acronym and a reference of this protocol to support this information.

-  Page 2, lines 78.80. I have doubts about the hypothesis of your study. In the abstract you mention that the objective of this study was to find out the preventive effects of the applied therapies (hot and cold), however according to your hypothesis you did not expect to obtain these effects. Could you please describe this in more detail to clarify it?

Material and methods

- Page 2, lines 83-88: Why did you add only male participants to the study, was there any reason not to add females? What age range did you determine and why? What inclusion or exclusion criteria were established to select the participants in the study? Please complete this information

- Page 3, line 100: What method or system did you use to randomize the participants in each of the groups? You mentioned that the examiners were blinded, so were the participants or professionals that applied the therapies blinded? Please add this information.

- Page 3, lines 101-106: This information refers to the results of your study, I suggest you to place this information in the results section

Page 3, line 112: Please, could you add information about the period of time and places in which the data were recorded (i.e.: university,…)? How long did it take to complete the entire assessment? The duration of the sessions would be different, since in the first session the participants perform the exercises and 30 minutes of treatment or rest are applied to the control group. What were the approximate time of the first session and the rest of the sessions?

- Pages 3-5, lines 137-221: You describe in detail the eccentric exercise-induced muscle damage protocol and each of the measurements carried out, however, you do not mention any reference in this entire paragraph except in muscle soreness and PPT. I assume that you have consulted previous bibliography and protocols to determine these measurements, please add references to support each one of them.

- Page 5, lines 224-233: Following the previous comment, I suggest that you add references that support the information about the protocols described for both applying interventions (heat and cold), this information is very important to your study.

- Page 5, lines 234-2389, figure 2. This information is very interesting. I suggest you mention in this section the pilot study and the measurements made of the temperature. However, I suggest that the data provided and figure 2 should be in the results section.

Results

-Page 6, line 254: I suggest you add some information from the participants in this section. For example, you can show the average age, anthropometric characteristics (height, weight) of each of the groups. Also if you have obtained any additional information from the participants such as level of studies, physical exercise that they usually practice, etc.

Page 6, line 264.268: It is confusing that figure 4B is mentioned in the text before figure 4A. I suggest changing the order. Put figure 4B first.

-Pages 6-10. In the text of the results: The interpretation of the results becomes difficult to follow at some points.

You mention for example: "Post hoc tests revealed significant decreases (P < 0.05) on days 1, 2 and 3 compared to pre. Furthermore, a significant increase was observed on day 7 compared to days 1, 2, 3 and 4".

What groups are you referring to specifically? The control group versus the heat group, the control group versus the cold group? The control group versus the two intervention groups?.. This is repeated throughout all the text that describes the results of the different figures. In the graphs this information cannot be appreciated with precision. You should therefore include this information throughout the text to help readers interpret and understand your results.

-Page 10. Figures. Please add information in the text and an explanatory caption on the photographs that appear accompanying figure 7 on page 10

Discussion

-Page 12, lines 446-456: You adequately mention the main limitations of the study. I suggest the following that could also be added: you have only included men in your study, which could represent a gender bias. A single intervention session is performed with heat or cold makes, so it is necessary future research to evaluate the effects of multiple sessions over a longer period. Although your sample size calculations ensured that the study was sufficiently powered; further studies with larger sample sizes are needed to corroborate your findings

-Page 12. Lines 458-463. Taking into account the limitations mentioned the results of your study should thus be interpreted with caution.  Therefore and consequently I suggest you to the reformulate your conclusions in a more carefully way such as: “Our results suggest that a 30-min heat or cold therapy administered immediately after DOMS induction in the elbow flexor muscles seems not to suppress….”

The second part of your conclusions is not in line either with your hypothesis or with the results that your study has shown.. “Because improvement of various symptoms caused by DOMS can be expected to be applied to rehabilitation and sports, both heat and cold therapies should be considered…..” Rewrite this sentence to match your hypothesis, results and expectations for the future.

I hope that my comments could help to improve the paper.

Reviewer 2 Report

Thank you for this interesting paper. I added a small comment in the review (word attached).

Reviewer 3 Report

This is a well-designed study investigated the DOMS symptoms along with a few muscle parameters after eccentric contraction (ECC), and also evaluated the effect of heat or cold intervention on preventing the ECC induced DOMS. Overall, this study appears to have been rigorously conducted and presented, and I have a few concerns as follows:

1.     This is a major problem. After reading through the whole manuscript, I then realized that this is a single bout treatment of heat or cold, so all conclusions are based on this one single 30 min treatment. If there is no effect, it does not necessarily mean that heat or cold treatment did not work, it might be the inadequate treatment with only one bout. Therefore, the title should be clarified a bit more, i.e. effect of single bout treatment of heat or cold.

2.     Follow up with the first question. Have the Authors ever tested a prolonged or repeated treatment with heat or cold? If so, it would be more comprehensive to understand the effect of heat or cold treatment on DOMS.

3.     For the treatment, the Authors only mentioned microwave therapy and ice cube for the heat and cold, any specified temperature for the treatment? i.e. 50 and 0 degree?

4.     In Figure 3, it only shows the total eccentric torque, is there any data showing the specific torque after normalizing the torque to the muscle size? As the muscle size would vary dramatically from people to people, and even different arms in the same people would show quite different strength, so specific torque or muscle force would be more significant than the total force or torque.

5.     Based on the current results, actually HEAT treatment has a strong trend to alleviate DOMS symptoms, although it might not be quite significant, any discussion on that? Or why HEAT treatment showed a better trend comparing to ICE treatment? i.e. muscle soreness in Figure 4?

6.      In Figure 4 panel B, the legend “HRAT” should be “HEAT”.

Round 2

Reviewer 1 Report

healthcare-2080973_review_R2

Title (New): The effect of single bout treatment of heat or cold intervention on delayed onset muscle soreness induced by eccentric contraction

Comments and Suggestions for Authors

Dear authors,

I was glad to have the opportunity to review the new version of your manuscript, you investigated the effects of a single session of heat or cold therapy applied on the elbow flexor muscles after repeated eccentric contraction on torque reduction, muscle soreness, and range of motion due to delayed onset muscle soreness in healthy male participants.

In my opinion, you have responded positively to the suggestions for improvement made, you have expanded the information required in the introduction and material and methods sections, you have made the modifications indicated in the results section, and reformulated the hypothesis and conclusions.

I believe that all these modifications have improved the quality of this manuscript.

I therefore consider that the current improved version can be accepted for publication.

I congratulate you on your great effort and the work you have done.